# Characterization of Vegetation Dynamics on Linear Features Using Airborne Laser Scanning and Ensemble Learning

**Narimene Braham** [1,*,†] **, Osvaldo Valeria** [1,2] **and Louis Imbeau** [1]

1 Institut de Recherche sur les Forêts, Université du Québec en Abitibi Témiscamingue, 445 Boulevard de l'Université, Rouyn-Noranda, QC J9X 5E4, Canada

2 Hémera Centro de Observación de la Tierra, Escuela de Ingeniería Forestal, Facultad de Ciencias, Universidad Mayor, Camino La Pirámide, Santiago 5750, Chile

* Correspondence: narimene.braham@uqat.ca

† This manuscript is part of a M.Sc. thesis by the first author, available online at (https://depositum.uqat.ca/).

**Abstract:** Linear feature networks are the roads, trails, pipelines, and seismic lines developed throughout many commercial boreal forests. These linear features, while providing access for industrial, recreational, silvicultural, and fire management operations, also have environmental implications which involve both the active and non-active portions of the network. Management of the existing linear feature networks across boreal forests would lead to the optimization of maintenance and construction costs as well as the minimization of the cumulative environmental effects of the anthropogenic linear footprint. Remote sensing data and predictive modelling are valuable support tools for the multi-level management of this network by providing accurate and detailed quantitative information aiming to assess linear feature conditions (e.g., deterioration and vegetation characteristic dynamics). However, the potential of remote sensing datasets to improve knowledge of fine-scale vegetation characteristic dynamics within forest roads has not been fully explored. This study investigated the use of high-spatial resolution (1 m), airborne LiDAR, terrain, climatic, and field survey data, aiming to provide information on vegetation characteristic dynamics within forest roads by (i) developing a predictive model for the characterization of the LiDAR-CHM vegetation cover dynamic (response metric) and (ii) investigating causal factors driving the vegetation cover dynamic using LiDAR (topography: slope, TWI, hillshade, and orientation), Sentinel-2 optical imagery (NDVI), climate databases (sunlight and wind speed), and field inventory (clearing width and years post-clearing). For these purposes, we evaluated and compared the performance of ordinary least squares (OLS) and machine learning (ML) regression approaches commonly used in ecological modelling—*multiple linear regression (mlr), multivariate adaptive regression splines (mars), generalized additive model (gam), k-nearest neighbors (knn), gradient boosting machines (gbm),* and *random forests (rf)*. We validated our models' results using an error metric—root mean square error (RMSE)—and a goodness-of-fit metric—coefficient of determination ($R^2$). The predictions were tested using stratified cross-validation and were validated against an independent dataset. Our findings revealed that the *rf* model showed the most accurate results (cross-validation: $R^2$ = 0.69, RMSE = 18.69%, validation against an independent dataset: $R^2$ = 0.62, RMSE = 20.29%). The most informative factors were clearing width, which had the strongest negative effect, suggesting the underlying influence of disturbance legacies, and years post-clearing, which had a positive effect on the vegetation cover dynamic. Our long-term predictions suggest that a timeframe of no less than 20 years is expected for both wide- and narrow-width roads to exhibit ~50% and ~80% vegetation cover, respectively. This study has improved our understanding of fine-scale vegetation dynamics around forest roads, both qualitatively and quantitatively. The information from the predictive model is useful for both the short- and long-term management of the existing network. Furthermore, the study demonstrates that spatially explicit models using LiDAR data are reliable tools for assessing vegetation dynamics around forest roads. It provides avenues for further research and the potential to integrate this quantitative approach with other linear feature studies. An improved knowledge of vegetation dynamic patterns on linear features can help support sustainable forest management.

**Keywords:** airborne light detection and ranging LiDAR; linear features; forest roads; road network; forest management; random forests; ensemble learning; vegetation dynamics; boreal forest

## 1. Introduction

Anthropogenic linear features are forest access infrastructure, namely forest roads and seismic lines, and are essential for boreal forest natural resource provisioning and transportation. These features may have distinct morphological characteristics and functions, but their similar geometry and spatial patterns result in analogous environmental effects which allow their approximation. Particularly, linear features (LFs) are similar in terms of their disturbance legacies as they require the use of machines that result in the compaction of the surface layer through construction operations and consistent traffic intensity [1–6]. A consequence of these legacies is prolonged post-clearing vegetation growth. LFs also play a major role in expanding forest cover discontinuity as they represent an extensive crisscross in terms of their spatial distribution. In terms of their geometry, LFs have higher perimeter-to-area ratios and higher edge-to-area ratios [7,8]. Even if some of these LFs are temporary or deemed to have a "low-impact" [9,10], they contribute to fragmentation, with the majority (70%) of the world's forests being within 1 km of a forest edge, leading to diminished habitat suitability adjacent to LFs caused by edge effects [11,12]. Moreover, LFs have direct effects on wildlife species [11,13–16], soil [5,17–19], seed dispersal and the spread of wind-dispersed invasive species [20], abiotic conditions [21,22], forest structure and composition, and their adjacent environment [23,24]. Since the most prevalent linear anthropogenic feature in many regions of eastern boreal forest are forest roads, the management of this vast network to minimize the associated linear footprint on biodiversity and wildlife habitat, requires an understanding of forest road vegetation characteristic dynamics [25,26]. However, vegetation patterns around forest roads need to be further explored: previous studies have shown that the growth process around linear features is complex and slow [27,28]. Furthermore, fine-scale knowledge on growth mechanisms around forest roads and the application of this knowledge to management of the linear footprint is based on limited spatial levels and time scales. Previous studies assessing the post-clearing, forest canopy spatio-temporal dynamic showed that the growth process is conditioned by disturbance factors, site conditions, and location [29,30]. Moreover, in natural canopy openings, factors such as light, nutrients, and water, have been shown to contribute and interact to affect the growth of individual trees and saplings [31]. Abib et al. [1] and Franklin et al. [21] confirmed this relationship for LFs and showed that variations in vegetation growth are explained by LF attributes (i.e., LF width and orientation), local environmental factors (i.e., sunlight availability and the potential for the accumulation of surface water) as well as terrain conditions. However, vegetation dynamics around forest roads require more research for a better understanding of the conditioning factors. The analysis of vegetation characteristic dynamics can be challenging if in situ measurements are used to acquire the information needed because forest roads are extensive throughout the landscape and have variable clearing widths which are permanently fluctuating over time due to vegetation growth in the immediate surroundings. Moreover, in situ measurements are restricted to a limited number of data points (high precision measurements from a few small plots) instead of continuous data, and require additional human resources to perform the field surveys. For this task, up-to-date, spatially explicit, and continuous information about vegetation three-dimensional characteristics (e.g., height and cover of the trees and shrubs, presence or absence of strata, canopy closure, gap fraction) is essential [32,33]. Remote sensing techniques can reliably expand the measurement possibilities of vegetation characteristics, across multiple levels (e.g., plot, landscape, region) and multiple time intervals. Particularly, LiDAR data can be used to accurately quantify a variety of metrics describing vegetation [34] as well as subcanopy topography [35]. Coupled with the fact that this information can be derived across a range of spatial scales from fine (e.g.,~1 m$^2$)

to coarse (e.g.,~100 km$^2$) [36], the use of LiDAR data should provide a way to advance the high-resolution quantification of vegetation and terrain characteristics around forest roads. For instance, high-resolution LiDAR data, in conjunction with various sources of ancillary data, have been recently incorporated into the modelling of fine-scale forest road deterioration [37,38]. LiDAR structural metrics related to height, density, and complexity are relevant for research on forest structural characteristics [39]. In our study, we considered a density-related LiDAR-Canopy Height Model (CHM) metric to derive the percentage of vegetation returns with a $\geq$1.3 height threshold [40]. This metric provides a measurement of the road surface covered in vegetation. The potential factors conditioning the vegetation cover response were selected to be available across the study area, consistent with the spatial resolution of available LiDAR data, and with the published literature assessing the influence on vegetation dynamics. In particular, the size of canopy openings [29,41], years post-clearing, disturbance history [29,30], topography and climate [42] were the main factors that have been shown to influence forest structural characteristic dynamics. Forests' structural characteristics are also determined by site conditions [43–46], species composition [47], and successional status [48]. Previous studies showed that these afore-mentioned candidate factors are relevant for the characterization of vegetation cover on LFs [21,27,28,49–51]. The extraction of ecologically relevant information on forest road vegetation characteristics requires the processing of canopy height model (CHMs) data into suitable metrics such as height metrics (e.g., the mean and maximum percentiles of height) and density metrics (e.g., percentage of vegetation returns $\geq$ a given height threshold) [52,53]. These metrics are then used to develop products related to environmental modelling and forest management (i.e., a predictive model or a set of predictive models). For this purpose, machine learning (ML) approaches are usually the selected tool in forestry applications in the form of both classification and regression tasks due to the absence of distributional assumptions and the ability to fit nonlinear and complex relationships characterizing environmental and ecological data. Examples of predictive approaches include nearest neighbor methods (*knn*), e.g., [54–58], and multivariate adaptive regression splines (*mars*), e.g., [59–61]. In particular, ensembles approaches, e.g., *gradient boosting machines* (*gbm*) and *random Forests* (*rf*), are the tools of choice in forestry [62–65], and in forestry modelling applications with airborne LiDAR [1,66,67]. The widely used *rf* tree-based ensemble approach [68] is based on an aggregation of decision trees and uses several methods to introduce added randomness: i) through resampling, i.e., each tree is grown on a subset of the training points, and ii) through factor restriction (i.e., each decision tree uses a randomly selected subset of both the available factors and observations). At each step of the decision tree building process, a subset of the factors is randomly chosen, and the best factor and split point is chosen from that reduced set of factors. The average of decision trees is used to predict new observations. Other characteristics of *rf* are the reduced number of parameters to calibrate, and the choice of these parameters generally having very little influence on the accuracy of the results [69]. Although the *rf* approach is sufficiently versatile and widely used for such modeling, often the predictive capability of other ML techniques is not explored. The *gbm* is another tree-based ensemble approach and is a recent advance in predictive modeling [70]. The decision trees are sequentially built from the residuals of the preceding tree(s) and iteratively perform boosting through choosing, at each step, an arbitrary sample of the data, ultimately causing a progressive improvement in the model's performance [71,72]. However, *gbm* has yet to be tested to predict vegetation characteristics. To optimize accuracy and avoid overfitting using ML approaches, model parameter specifications are an important step. They usually involves a number of interacting parameters that have to be calibrated (i.e., regularized) in order to achieve optimal results [73]. Our primary aim is: (i) to investigate the predictive performance of six modelling approaches (*mlr*, *gam*, *mars*, *knn*, *rf*, and *gbm*) for the characterization of within-forest road vegetation cover dynamics, and (ii) to provide information on the underlying factors conditioning vegetation cover dynamics. We assumed that machine learning (ML) approaches would have better accuracies than ordinary least squares (OLS)

approaches. More specifically, tree-based approaches would show improved vegetation cover predictions. The evaluated approaches were constructed using ancillary geoclimatic as well as field inventory data. The required parameters for model fitting were set by using a 10-fold stratified cross-validation with 20 repetitions. For the final fitted model, parameters with the lowest error metric (root mean square error) were used and accuracy measures and analyses were conducted using both cross-validation and an independent validation dataset. The combined use of LiDAR measurements and predictive modelling allow for a fine-scale and representative measurement of forest road vegetation cover dynamics. This would also enhance our ability to precisely predict this dynamic along a spatial continuum and over extended timeframes.

## 2. Study Sites

For this study, we retrieved forest road clearing width data from the field, across three study areas representative of Canadian forestry activity, between 47 and 49° N and 72 and 78° W, in the mixed and coniferous boreal forest of Quebec (Canada) (Figure 1). The field data were collected in August 2019, as described in Girardin et al. [37]. The climate across our study areas is typically boreal, with very cold winters and short cool summers. The temperatures change according to latitude and altitude, with the southernmost and northernmost sites being the warmest and the coldest, respectively, and the sites at higher altitudes being the coldest in winter and the least warm in summer. Precipitation also varies along the latitudinal gradient, with drier conditions toward the North. The mean annual temperatures range between −5.9 and 4.2 °C and total precipitation ranges between 650 to 1424 mm. The May–September mean temperatures range between 9.1 and 17.7 °C. The study areas are characterized by a gently rolling topography, with the highest mountains concentrated in the southern part, and thick and undifferentiated glacial deposits [37,74–76]. Table 1 provides a brief description of our study sites.

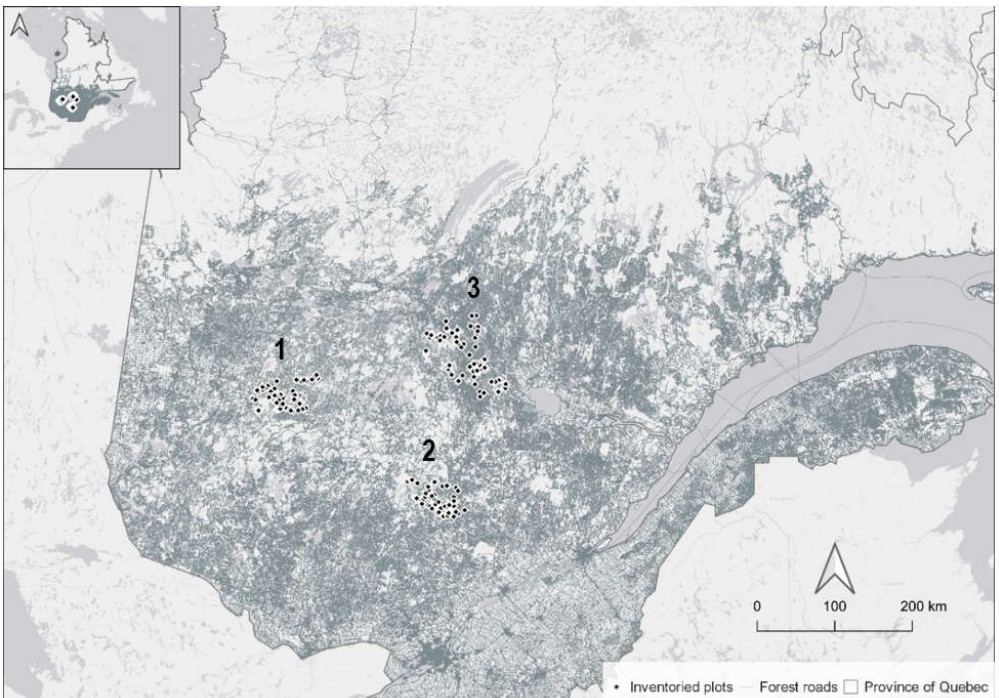

**Figure 1.** Overview of forest road network (dark grey polylines) and distribution of sampled field plots (black dots) within the three respective study areas (1−3) in the province of Quebec in eastern Canada.

**Table 1.** Properties of forest roads and their bioclimatic data, grouped by study area (1−3). The information in the table is in part adapted from [37,76–78].

| Characteristic | Study Area 1 | Study Area 2 | Study Area 3 |
|---|---|---|---|
| Location | Northeastern Abitibi-Témiscamingue region | Mauricie region | Northeast of the Saguenay-Lac-Saint-Jean region |
| Latitude/Longitude | (48.42° N, 77.23° W) | (47.51° N, 72.78° W) | (48.89° N, 72.23° W) |
| Mean elevation of sampled roads (m) | 393 | 430 | 407 |
| Total number of sampled plots | 84 | 73 | 84 |
| Cumulative length of sampled roads (km) | 4.2 | 3.65 | 4.2 |
| Mean clearing width measured in the field (m) | 8.59 | 7.74 | 8.55 |
| Mean years post-clearing (years) | 9.23 | 6.83 | 6.17 |
| Mean slope (%) | 5.10 | 5.58 | 4.27 |
| On-road mean vegetation coverage * measured in the field (m) | 0.47 | 0.41 | 0.46 |
| On-road mean tree height measured in the field (m) | 4.22 | 6.08 | 5.22 |
| On-road mean shrub height measured in the field (m) | 1.24 | 2.87 | 2.19 |
| Average annual temperature (°C) | 1.5 | 3.8 | 1 |
| Annual precipitation (mm) | 875 | 928 | 999 |
| Bioclimatic domain/Vegetation type | Balsam fir [*Abies balsamea* (L.) Mill.]—White birch (Betula papyrifera Marsh.) | Balsam fir—Yellow birch (Betula alleghaniensis Britton) | Black spruce Picea mariana (Mill.)—Moss domain and Balsam fir—White birch |

* Vegetation coverage measured as the ratio of the mean width of the road covered in vegetation to the original width of the road, both measured in the field.

## 3. Data

### 3.1. Reference Data

We used 240 rectangular field plots (50 m length) which were at least 250 m apart from one another. These field plots were randomly sampled among a selection of forest road stratified by the clearing width class (three classes: narrow, medium, and wide), years post-clearing (YPC) class (two classes: short-term and long-term timeframes), and slope class (two classes: low and high longitudinal slope, range: 0%–16%), following Girardin et al. [37]. Clearing width varied between 4 and 14.4 m and included winter only roads and all-weather gravel roads. Paved highways were not considered. YPC ranged between 0 and 46 years and was estimated based on the time elapsed since the last clearing (maintenance or construction). Maintenance activities usually consist of culvert repairs, surfacing, layer gravelling, and vegetation clearing. These reference data were used for the retrieval of geospatial information spanning from the road centerline, as described in the data extraction step (Section 3.2) and Figure 2.

For visualization purposes, clearing widths were binned into narrow forest roads (total narrow forest roads = 96), which were ≤7 m wide, and wide forest roads, which were >7 m wide (total wide forest roads = 144) [6].

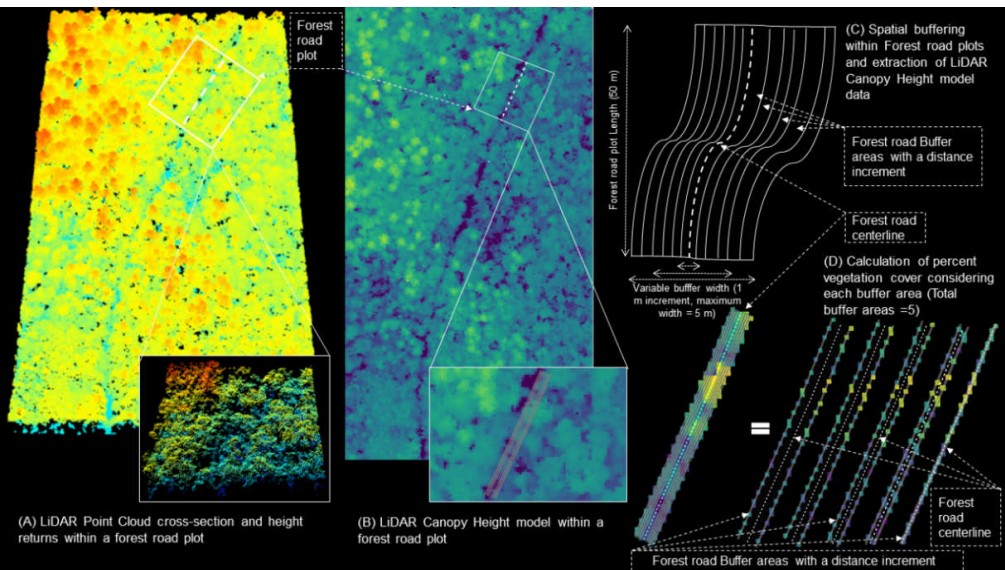

**Figure 2.** Visualization of LiDAR-based data. (**A**) 3D point cloud. (**B**) Canopy height model (CHM) over a forest road. (**C**) Extraction of forest road plot-level vegetation cover (%) using the CHM. (**D**) Calculation of mean vegetation cover, continuously, within the five multi-buffer areas (length = 50 m, and width increment = 1 m).

### *3.2. Forest Road Data Extraction*

#### 3.2.1. Digital Surface Models (DSM) and Canopy Height Models (CHM)

LiDAR point clouds feature the heights of objects on the ground. Digital surface models (DSM), canopy height models (CHM), and digital terrain models (DTM) are common layers derived from point clouds after the classification of individual LiDAR points. DSM and CHM feature the highest elevation of ALS returns. DTM represents the elevation of the ground. In vegetated areas, the CHM represents the heights of the trees on the ground. It can be derived by subtracting the ground elevation (represented by the DTM) from the elevation of the top of the surface, or the tops of the trees (represented by the DSM). Different methods exist to create gridded DSMs and CHMs [79]. In this study, we used LiDAR-based gridded products provided by the government of Quebec [80].

#### 3.2.2. Spatial Buffering for Data Extraction

To recreate the footprint polygons of forest roads from field-inventoried centerlines, we first delineated and digitized the centerlines using GPS coordinates (Trimble GNSS Handheld Geo7X, provided by Trimble Inc., Wesminster, CO, USA) (three sampling locations for the edges and midpoint of the 50 m centerline) (Supplementary Material, Figure S1). To ensure the proper alignment of the digitized centerlines, we used the LiDAR datasets provided by the Government of Quebec's airborne LiDAR surveys, consisting of 1 m × 1 m grids [80], collected under growing season conditions between 2016 and 2020, with a mean pulse density of 2−4 pulse/m² [81]. More specifically, we derived the topographic position index (TPI) from the digital terrain model (DTM) (spatial resolution of 1 m) to locate topographic breaks and inspect roadside geomorphological attributes (i.e., the drainage structures or ditches). We then performed a buffer analysis to partition the geographic space around the digitized centerlines into multi-buffers with similar areas (1 m increment). This spatial buffering step resulted in 1 m wide "hollow" multi-buffers that extend over 5 m, which we used to compute our input dataset (vegetation cover response and causal factors) for the characterization of the vegetation dynamics (Figure 2). All data processing, modelling, as well as validation were performed in the R project for statistical computing, software environment (Version 4.1, R Core Team) [82]. All regression models were produced using the *caret* library 6.0 [83].

### 3.2.3. Causal Factor Data Computation

We established a framework that used data from multiple sources, including airborne LiDAR and geo-climatic data. We extracted these data from the 240 field plots using the multi-buffer delineation approach described in Section 3.2.1 and Figure 2. The proposed approach had a fixed length (50 m) and a variable width extending over 5 m, which allowed us to derive our data with a distance increment from the road centerline. All training data were extracted within the boundaries of the delineated multi-buffer areas, annotated 1 to 5, indicating the buffer width. For buffer areas more than one meter wide, data were extracted within hollow bands to exclude data points from the other buffers.

Specifically, we used the LiDAR-CHM data to measure the vegetation cover response and LiDAR-based terrain data (1 m resolution) to compute: (i) Slope, in degrees. (ii) orientation (northernness) transformed to a continuous factor ranging between $-1$ and $1$ (The northernness values closer to $-1$ are southwards and those closer to $+1$ are northwards) [28]. Orientation is typically transformed into a continuous factor because it is circular (large values may be very close to small values). (iii) The topographic wetness index (TWI) is used as a proxy for soil moisture. It provides information on the potential for water accumulation over the land as a function of slope and accumulation at a given pixel. More specifically, TWI integrates the water supply from an upslope catchment area and downslope water drainage for each cell in a digital terrain model [84]. (iv) Hillshade is a proxy for the shadow based on the surface elevation [85,86].

NDVI (normalized difference vegetation index) extracted from Sentinel-2, resampled to 1 m resolution, provides a measure of the difference between the reflectance of wavelengths emitted by the sunlight in the near infrared (PIR) and in the visible red band [87,88].

Climate data were obtained from WorldClim (Version 2.1) for the time period 1970–2000 [89,90]. This dataset is based on historical climate records at a resolution of 30 s. The available monthly climate data of precipitation, incident sunlight (in units of $kj \cdot m^{-2} \cdot day^{-1}$ wind speed ($m \cdot s^{-1}$), total precipitation (mm), and minimum, mean, and maximum temperature ($^{\circ}C$), were used to compute the growing season climate dataset, resampled to a 1 m resolution. Only two growing season averaged climatic factors, namely incident sunlight and wind speed, were retained for further analyzes, because a high correlation between the initial variables was found in Pradhan and Setyawan, 2021 [91]. Particularly, sunlight is a proxy for vegetation growth as it moderates the available photosynthetically active radiation. Sunlight and wind speed are proxies for the potential for in situ evapotranspiration due to locally warmer/drier or cooler/shaded conditions, as suggested in Stern et al. [22], and van Rensen et al. [28].

Prior to the modelling analysis, we checked for outliers using the interquartile range and removed all values above the 95th and below the 5th percentile, as well as collinearity (relationships between more than two covariates), and correlation (linear relationships between two covariates), following Zuur et al. [92]. All uninformative metrics that showed a variance inflation factor greater than 3 or were highly correlated with one another ($|r\ Pearson| > 0.7$) were excluded from the analysis. We summarize in Table 2, the various factors examined and their description.

**Table 2.** Overview of the factors used in the modelling of vegetation cover. Geospatial layers had a cell resolution of 1 m or were resampled to 1 m prior to the modelling step, for all the factors.

| Data Source(s) | Factor(s) | Unit | Description | Spatial/Temporal Resolution | |
|---|---|---|---|---|---|
| LiDAR-based, CHM | Vegetation cover (response) | % | Mean vegetation cover (height above 1.3 m) within the buffer area | 1 m | - |

**Table 2.** *Cont.*

| Data Source(s) | Factor(s) | Unit | Description | Spatial/Temporal Resolution | |
|---|---|---|---|---|---|
| LiDAR-based, Terrain | (i) Slope | % | Mean slope within the buffer area | 1 m | - |
| | (ii) Orientation (Northernness) | Unitless | Mean northernness index within the buffer area | 1 m | - |
| | (iii) TWI | Unitless | Mean TWI index within the buffer area | 1 m | |
| | (iv) Hillshade | Unitless | Mean hillshade index within the buffer area | 1 m | |
| | NDVI | Unitless | Mean NDVI index within the buffer area | 1 m | |
| Climate | Solar radiation | $Kj \cdot m^{-2} \cdot day^{-1}$ | Mean solar radiation within the buffer area | 1 m | 30 s |
| | Wind speed | $m \cdot s^{-1}$ | Mean wind speed within the buffer area | 1 m | 30 s |
| Linear feature attributes | Clearing width | m | Line width derived from three measurement plots along the 50 m plot | | - |
| | Years since last clearing (clearing) (YSC) | years | Time since last clearing (establishment or maintenance) | - | - |

## 4. Methods

### 4.1. Statistical Approaches

To provide an optimal predictive model for the estimation of vegetation cover on forest roads, we compared the performance of the following OLS regression approaches: *(i) multiple linear regression (mlr), (ii) multivariate adaptive regression splines (mars), (iii) generalized additive model (gam), (iv) k-nearest neighbors (knn), (v) random forests (rf), and (vi) gradient boosting machines (gbm).*

*mlr* was assessed for its straightforwardness and simplicity and was extended to *gam*, a flexible approach used to identify and characterize non-linear regression effects [93].

*gam* was included because it presents an advantage over predefined basis functions to achieve nonlinearities and is relatively easy to interpret [94].

The parsimony of *mlr* and *gam* approaches were assessed with the Akaike information criterion (AIC) [95]. All possible combinations of factors and interaction effects were analyzed with the *MuMIn* library in R [96]. This step was essential because the inclusion of uninformative factors in parametric and semi-parametric models (i.e., *mlr* and *gam*) can reduce their overall predictive performance.

*mars* is also regarded as an extension of linear models and is an adaptive non-linear estimation method that can present interaction between influencing attributes without any assumptions about input data distribution [97]. It structures a relation from established basis functions and coefficients, which are generally determined from the regression information [98]. The construction phase of a *mars* model involves adding and removing of basic functions. *mars* is considered as a modification of the classification and regression tree (CART) method, to improve the latter's performance in a regression setting, owing to *mars'* ability to capture additive effects [93]. Therefore, *mars* could simplify the challenges of solving non-linear relationships, compared to other non-parametric approaches [98].

We used the basic *knn* method [99], a simple and intuitive approach in which each observation is predicted based on its similarity to other observations [69]. More specifically, the prediction of new observations values uses the sampled observations from a training

data set that are the closest (nearest neighbor(s)) to each new observation. The similarity between new and training samples is based on a Euclidean distance metrics (or other related metrics) [100]. *knn* is considered a simple approach as there is no model to be fit and the prediction results depend on feature scaling, measurement of similarity, and the value of k. Other advantages include decent predictive power, especially when the response is dependent on the local structure of the features [100], flexible assumptions regarding normality and homoscedasticity required by parametric methods, and the preservation of much of the covariance structure among the metrics that define the response and factors' vectors [99].

*rf* is tree-based ensemble which builds a large collection of independent decision trees to further improve predictive performance by averaging individual predictions. More specifically, *rf*s use a combination of bagging, which randomly selects factors with replacement as training for growing the trees, which makes it robust against overfitting [101]. The training is carried out on datasets created from a random resampling on the training set itself, which adds an extra layer of randomness [68,101].

*gbm* is another recent tree-based ensemble which builds a base model (i.e., trees with only a few splits) [102] and the additional trees iteratively correct mistakes made by the previous trees, which progressively improves prediction accuracy. Particularly, *gbm* sequentially generates base models from a weighted version of the training data to find the optimal combination of trees and optimize predictive performance [69,103].

Both *rf* and *gbm* present the numerous advantages of tree-based ensemble methods, accommodating different types of factors and efficiently dealing with missing data and outliers. They have no need for prior data transformation, can fit complex non-linear hierarchical relationships, and automatically handle interaction effects between the factors [94].

### 4.2. Model Parameter Tuning

ML model performance can benefit significantly from tuning as it may reduce overfitting [73,104]. The *caret* library [83] was used to execute a grid search for each model where we assessed every combination of parameters of interest. More specifically, for *mars*, relevant model parameters were related to the number of retained terms (nprune) and the degree of interactions (degree) [69,105]. The implementation and performance of *knn* approaches required choices for three parameters: the value for k, the number of nearest neighbors (in a regression setting, for k = n, the average is used across all training samples as the predicted value), a scheme for weighting neighbors when calculating predictions (kernel function), and a similarity metric (distance). The prediction performance of *rf* is influenced mainly by three model parameters: correlation between individual trees, the performance of each tree, and the total number of trees [106]. Hence, we executed a grid search to evaluate: ntree, which is the number of trees in a forest, and mtry, which defines the number of random factors at each split [69]. For *gbm*, we performed sensitivity analyses on tree complexity (interaction depth), learning rate (shrinkage), and the minimum number of observations in nodes (minobs) [69,105]. During the tuning phase, a stratified 10-fold cross-validation resampling method allowed us to partition the training set for each fold. Model performances of every parameter combination were computed at the tuning level and averaged across all folds. The parameter combination with the lowest RMSE was used to train our model during the performance assessment phase. Details about the parameter values and combinations that optimized the RMSE for our data can be found in Supplementary Material Table S1.

### 4.3. Model Performance, Comparison, and Diagnostics Using Cross-Validation and Independent Dataset

Inheriting spatial information from dependent observations is one of the main challenges of spatial statistical modeling using ML techniques [73,107–110]. In this regard, to account for spatial dependencies in our spatially explicit data and reduce prediction bias, the choice of cross-validation (resampling technique) emerged as an important step in

the implementation of our approaches [73,109,111]. Therefore, we performed a stratified 10-fold cross-validation, with the forest road identifiers being the stratifying factor. This allowed the condition of equal distribution of our stratified samples between (i) training, testing, and validation samples, and (ii) the cross-validation folds to be met (e.g., [112]), which showed that dividing by strata produces similar distributions between training and testing sets for the majority of validation folds. The stratified partitioning was conducted prior to modelling and the samples were randomized with respect to the established strata. It is suggested that when the set of factors affect the response in different ways (positive/negative and/or linear/non-linear) and a model's output is transferred to unsampled locations, more rigorous validation is necessary [113]. We conducted a 60%–40% training–validation combination to evaluate our model's performance. In addition, to avoid skewed results, each model was run 20 times (20 repetitions). Both stratified cross-validation and independent validation (using the hold-out 40% of our data) performance were evaluated with the RMSE and the mean absolute error (MAE) metric to assess the accuracy. The $R^2$ metric was used to evaluate the goodness-of-fit. Model performance metrics were taken as the mean from the number of repeats. After the models were trained and compared, we assessed visual diagnostics and factor importance computed from the fitted model that yielded optimal results (i.e., *rf*). *rf* typically includes a permutation-based importance measure which assesses the decrease in accuracy averaged over all the trees for each factor. The factors with the largest average decrease in accuracy across all trees are considered the most important [69]. The factor importance computation was implemented using the *varImpPlot* function in the *Random Forest* library [114]. Partial dependence plots (PDP*s*) are especially useful for visualizing the relationships discovered from ML approaches by isolating the effect of a single factor on the response [115]. We evaluated the partial dependence from our fitted *rf* model using two functions *partial* and *plotpartial* [116] as there are advantages for model specific interpretations such as a close relation to the model performance and an accurate incorporation of the correlation structure between factors [115].

## 5. Results

### 5.1. Modelling Approaches' Performance

For the study, vegetation cover (LiDAR-measured vegetation cover (%)), forest road attributes (clearing width (m) and years post-clearing (years)) by means of in situ measurements, climatic factors (sunlight (kj·m$^{-2}$·day$^{-1}$) and wind speed (m·s$^{-1}$)), terrain factors (slope (%), northernness (index), TWI (index) and shade (index)) were computed. An overview and the distribution of these input data are summarized in Table 3.

**Table 3.** Distribution of model input data for the characterization of vegetation cover dynamics on forest roads.

| Input (s) | Min | Max | Range | Median | Mean | Standard Deviation |
|---|---|---|---|---|---|---|
| LiDAR measured vegetation cover (%) | 0 | 100 | 100 | 0 | 22.07 | 33.36 |
| Slope (%) | 0 | 27.73 | 27.73 | 6.71 | 7.94 | 5.41 |
| Northernness (index) | −0.55 | 0.46 | 1 | −0.01 | −0.03 | 0.2 |
| TWI (index) | 1.72 | 16.46 | 14.74 | 6.52 | 6.88 | 2.81 |
| Hillshade (index) | 139.68 | 202.97 | 63.29 | 178.82 | 177.7 | 9.85 |
| NDVI (index) | 0.12 | 0.89 | 0.77 | 0.66 | 0.62 | 0.19 |
| Sunlight (kj·m$^{-2}$·day$^{-1}$) | 17,228.74 | 17,729.8 | 501.06 | 17,598.99 | 17,545.63 | 136.74 |
| Wind Speed (m·s$^{-1}$) | 2.2 | 2.88 | 0.68 | 2.34 | 2.45 | 0.2 |
| Clearing width (m) | 4 | 14.47 | 10.47 | 7.4 | 8.24 | 2.48 |
| Years post-clearing (years) | 0 | 39 | 39 | 7 | 7.79 | 8.35 |

The predictive performance of ML approaches (*rf*, *gbm*, *knn*, and *mars*) and OLS (*gam* and *mlr*) approaches using stratified cross-validation and independent datasets are shown in Figure 3A,B, respectively. ML approaches consistently had higher testing and validation RMSE and higher $R^2$ values than OLS approaches. The greatest accuracy was obtained with the *rf* approach (RMSE ranging from 18.69% to 20.29% and $R^2$ ranging from 0.69 to 0.62), followed by *gbm* (RMSE ranging from 19.23% to 21.16% and $R^2$ ranging from 0.68 to 0.59), and finally *knn* (RMSE ranging from 21.59% to 21.73% and $R^2$ ranging from 0.59 to 0.56).

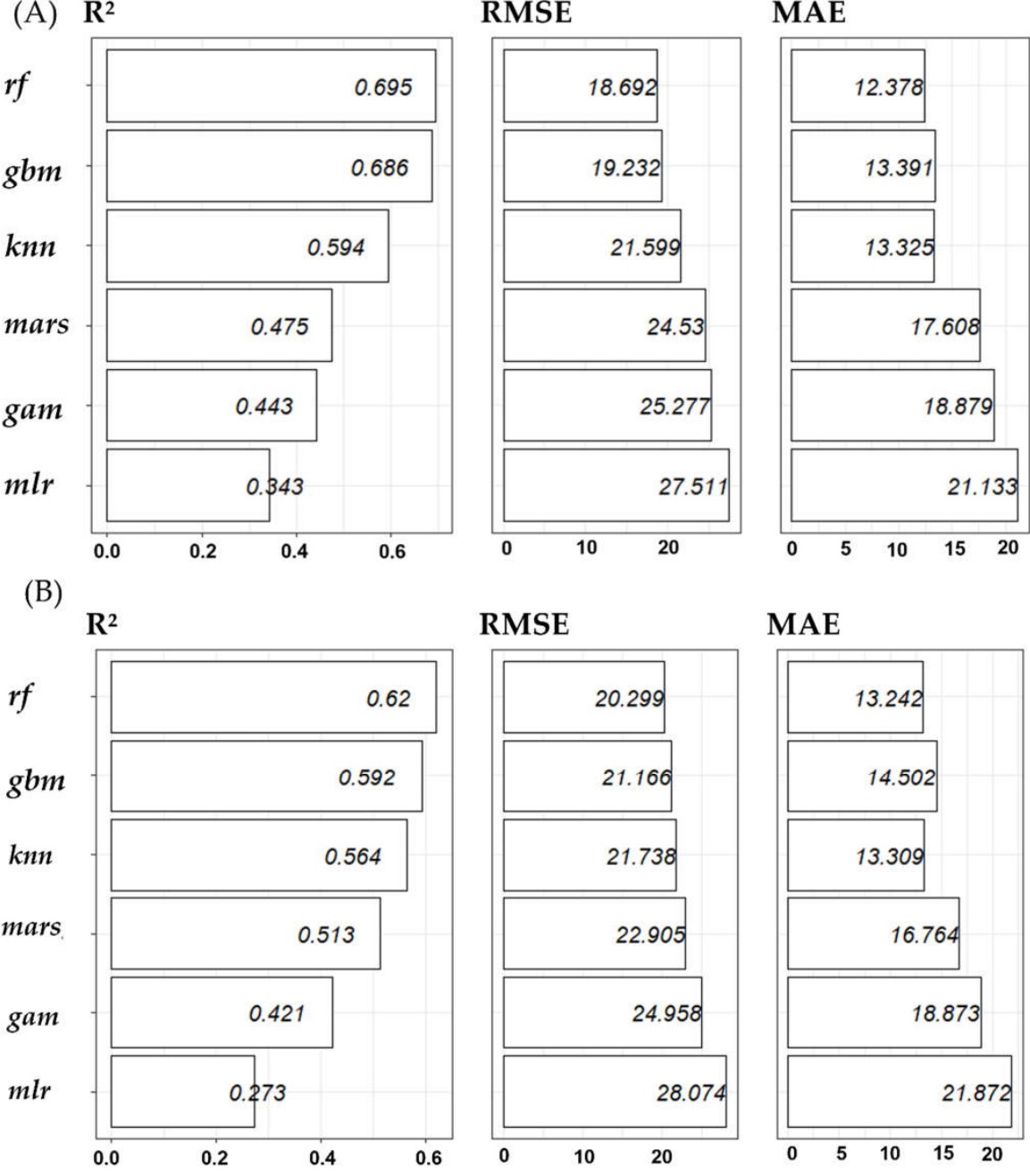

**Figure 3.** $R^2$, RMSE, and MAE for ML and OLS approaches for the characterization of vegetation cover dynamics obtained from (**A**) 10-fold stratified cross-validation (results from 20 repetitions were considered) and (**B**) an independent validation dataset. *rf = random forests, gbm = gradient boosting machines, knn = k-nearest-neighbors, mars = multivariate adaptive regression splines, gam = generalized additive model, mlr = multiple linear regression.*

Assessed using RMSE and R$^2$ (Figure 4A,B), the highest relative improvement in predictive performance was found using tree-based ensemble approaches (i.e., *rf* and *gbm*). Particularly, *rf* and *gbm* were similar in terms of predictive capability; they showed the highest predictive accuracy. *knn* and *mars* approaches showed slight reductions in the predictive capability compared with the *rf* and *gbm*, and significant reductions were obtained with the *mlr* approach compared with *rf*.

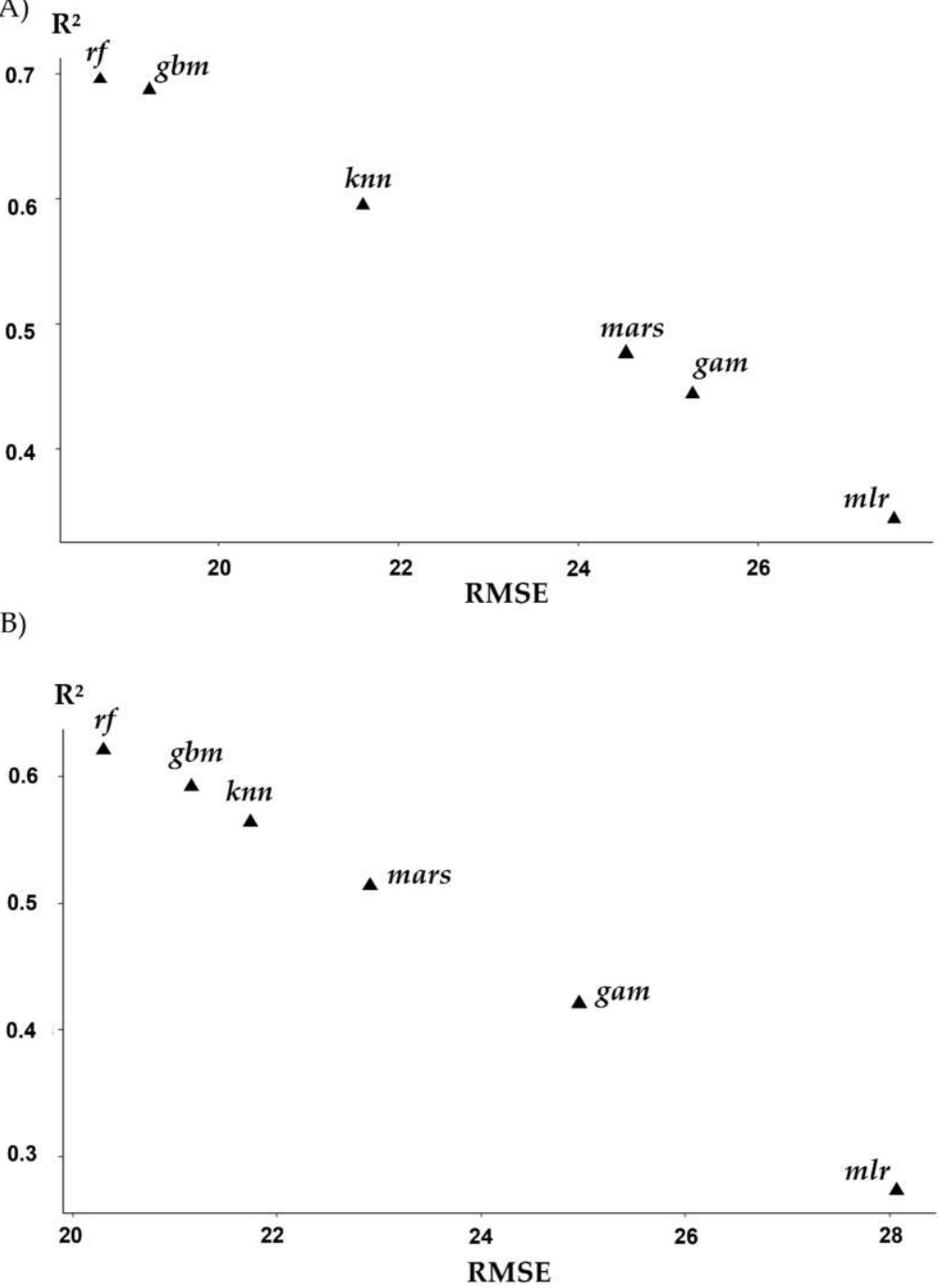

**Figure 4.** Predictive performance of ML and OLS for the characterization of vegetation cover dynamic using (**A**) 10-fold cross-validation approaches, (**B**) An independent validation dataset. *rf = random forests, gbm = gradient boosting machines, knn = k-nearest-neighbors, mars = multivariate adaptive regression splines, gam = generalized additive model, mlr = multiple linear regression.*

The causal factors which contributed most to the accuracy of vegetation cover characterization using *rf* are shown in Figure 5. Because *rf* generally provided optimal performance results, factor ranking was derived using this approach. Clearing width was the most important factor explaining vegetation cover dynamics around forest roads. The importance of all the other factors was lower: years post-clearing (YPC), NDVI, as well as geoclimatic (wind speed, sunlight, slope) and shade factors were of intermediate importance. The PDPs of the *rf* regression revealed a general downward trend of vegetation cover with increasing clearing width, sunlight, hillshade and TWI as well as a general upward trend with increasing years post-clearing, wind speed, slope, northernness and NDVI. PDPs for clearing width show that vegetation cover drops substantially as the clearing width increases until the width was approximately 6 m (Supplementary Material, Figure S3).

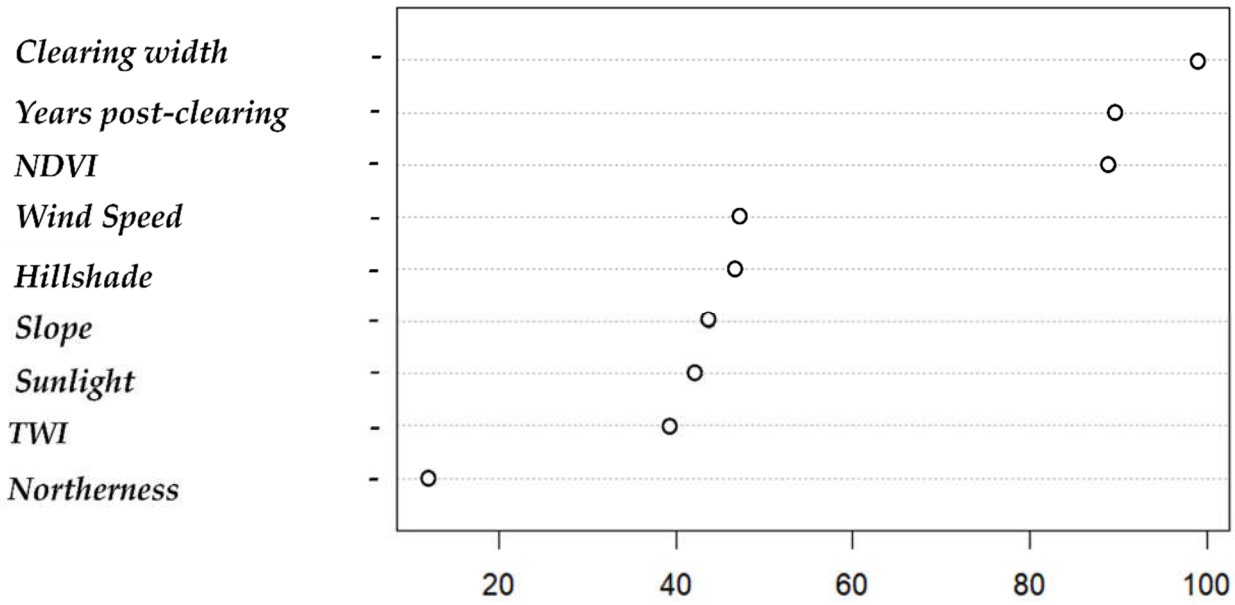

**Figure 5.** *rf*-based factor importance by permutation accuracy. A higher average importance of the variable (*X*-axis) indicates a greater contribution of this individual variable in explaining within-forest road vegetation cover dynamic. A ranking of all factors is included.

*5.2. Characterization of Vegetation Cover Dynamic around Forest Roads*

*rf*-based vegetation cover dynamics grouped by buffers extending from the road centerline (1–5 m), timeframe (short-, mid- and long-term), and clearing width (narrow and wide) are shown in Figure 6A using the cross-validation predictions, and Figure 6B using the independent dataset predictions. Overall, vegetation cover predictions were greater within the buffers furthest from the centerline. For the short-, mid- and long-term timeframes, the patterns were consistent: vegetation cover increased with YPC, with vegetation cover predictions on narrow forest roads slightly exceeding those on wide forest roads. Particularly, predictions grouped by timeframe showed that long-term vegetation cover (>20 YPC timeframe) exceeded those experienced in the mid- ([10–20] YPC timeframe) and short-term ([0–10] YPC timeframe), indicating a positive effect of YPC. Vegetation cover varied also across forest road types: narrow forest roads exhibited higher predictions over time across all five buffers with a higher range and higher mean predictions. The lowest prediction (~1.6%) was shown for wide roads for the short-term timeframe and the highest (~82.3%) for narrow forest roads for long-term timeframes. Wide forest roads showed an average vegetation cover of ~3%–53% and ~14%–52% in the mid- and long-term, respectively. Narrow forest roads showed an average of ~17%–51% and ~40%–82%, in the mid- and long-term, respectively (Supplementary Material, Figure S2A,B).

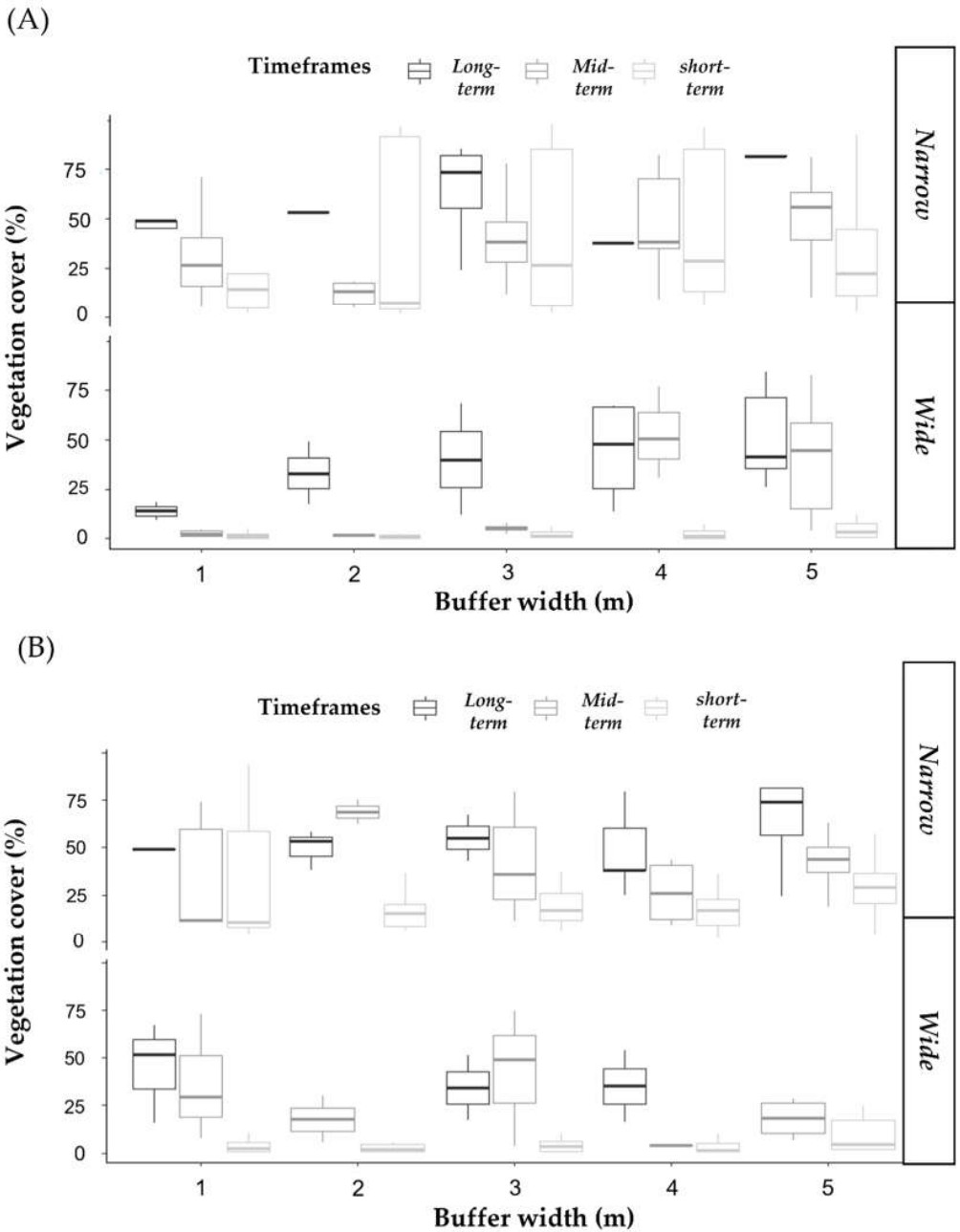

**Figure 6.** (**A**) Boxplots representing cross-validated *rf* model predictions ($R^2$ = 0.69, RMSE = 18.69%) of vegetation cover recorded within the multi-buffers extending from the road centerline, across forest road types (wide and narrow roads) for the post-clearing timeframes: >20 YPC (long-term, black boxes), [10–20] YPC (mid-term, dark grey boxes), and [0–10] YPC (short-term, light grey boxes). (**B**) Boxplot of vegetation cover predictions values from the *rf* model ($R^2$ = 0.62, RMSE = 20.29%) considering the independent validation dataset. The *X* axis indicates the width of every individual buffer. Boxplots present the median (dark black line), ±1 standard deviation (rectangle) and maximum-minimum value (vertical lines or whiskers).

As shown in Figure 3A,B, the stratified cross-validation testing dataset had a higher accuracy of prediction than the independent validation dataset. Both testing (cross-validation) and independent validation datasets were considered as stratified random samples, but the testing dataset had a closer relationship with the training dataset (reference population), as records from all strata were included in both the training and the test subsets (Figure 6A,B).

In general, we found the ML approaches evaluated here to be useful tools for improving predictions of vegetation cover dynamics on forest roads.

## 6. Discussion

### 6.1. Modelling Approaches' Performance

The performance results of OLS and ML approaches demonstrated that *rf* was the most reliable model, exhibiting the best prediction accuracy rates among the *gbm*, *knn*, *mars*, and *gam* approaches. The least accurate model was *mlr*. These results suggest that using ML approaches was appropriate for the characterization of vegetation cover dynamics around forest roads. Furthermore, compared to *rf* and *gbm*, *knn*, *mars*, and *gam* showed minimal accuracy reductions. Conversely, *mlr* performed poorly. The significant performance difference between *mlr* and *rf* can be explained by the limitation in handling non-linear relationships between the vegetation cover response and causal factors, as well as model assumptions about the non-linear distribution of input data: *rf* better accommodates nonlinear relationships between factors that *mlr* could not adequately solve [117–119]. Consistent with our hypothesis, tree-based ensemble approaches outperformed their nonensemble counterparts. *rf* and *gbm* are extremely randomized trees and are both based on ensemble learning theory. The ensemble—aggregation of decision trees [117]—considerably improves the accuracy and certainty of the predictions by suppressing the weaknesses and disadvantages of each individual decision tree, and by taking advantage of the responses of the combined decision trees [66,68,120,121]. ML approaches require the setting of parameter specifications prior to modeling to reduce overfitting and enhance performance. For this reason, the use of *rf* can be more straightforward because of its ability to yield accurate results when default parameters are used [122]. These findings, and previous results, suggest that no single ML algorithm might serve best for every task and that many models should be calibrated to identify the most accurate model for a given prediction task [55,104,113,118,123,124].

### 6.2. Factors Conditioning Vegetation Cover Dynamic around Forest Roads

#### 6.2.1. Factors Associated with Vegetation Dynamics

Our results identified that the most influential factors that explained significant vegetation cover variations were clearing width and years post-clearing (YPC). Particularly, vegetation cover was greatest in samples with narrow widths and long post-clearing time frames. NDVI, terrain (i.e., slope, hillshade, TWI, and northernness) and climatic factors (i.e., wind speed and sunlight) ranked lower. The samples where vegetation cover was most advanced had higher NDVI values, steeper slopes, higher orientation values, higher levels of wind speed, lower incident sunlight, shade, and TWI levels. Abib et al. [1], and Franklin et al. [21], showed that variations in proximity-based vegetation cover are explained by LF attributes (i.e., LF width and orientation) and local environmental factors (i.e., incident sunlight and the potential for accumulation of surface water). More evidence comes from van Rensen et al. [28], where clearing width was a strong predictor of growth occurrence within LFs (>3 m height cut-off was applied as a criterion for growth occurrence). It was suggested that clearing width implicitly reflects the severity of soil disturbance moisture supplies. Additionally, the ecosite type was the most important factor associated with growth (LF lines in bogs and fens were less likely to experience growth than those in drier conditions). Similarly, Finnegan et al. [125], suggested that soil wetness, nutrients, and adjacent stand affected growth levels. LFs in wet areas were least likely to promote vegetation growth and wet seismic LFs that were adjacent to more open forest stands were more likely to promote the occurrence of disturbance-tolerant taxa.

#### 6.2.2. Clearing Width and Its Relationship to Disturbance Legacies

Narrow-width forest roads experienced higher levels of vegetation cover, likely because of reduced disturbance (i.e., use of machinery in the construction phase and continuous vehicular traffic), supporting findings from the LF literature [27,28]. Particularly, LF

construction and design specifications can differ with respect to their characteristics (e.g., bearing capacity) and moisture conditions [126]. These differences are reflected in their trafficability, frequency, and intensity of use [127]. For instance, coarse material with higher levels of granular content (coarse gravel and/or crushed rock) is frequently used as a top layer on wide LFs to ensure higher bulk density and bearing capacity [127,128]. Due to their high trafficability, wide LFs are also prone to experience an increased intensity of use by heavy machinery (heavy vehicles inflict more damage to the surface layer than lighter vehicles), trucks, and off-road vehicles, which lead to severe disturbance of the top surface over longer time frames [6,38,127,129,130]. A consequence of compaction is the alteration of the hydro-physical properties in the surface layer. Therefore, it is likely that increased trafficability results in higher levels of compaction, which reduce porosity and infiltration, increase pore water pressure in the road material, and lead to long-term restricted water exchanges, flow, and moisture storage capacity. Gartzia-Bengoetxea et al. [131], showed that soil compaction caused by shearing and ripping persisted for 15 years. In addition, water holding capacity was lower in mechanically prepared plots 15 years after site preparation. Cambi et al. [132], showed that except for coarse textured excessively drained soils, soil compaction reduces oxygen and water availability to roots and microorganisms. Zang and Ding [51], suggested that compaction potentially interferes with the establishment of woody species on the surface of the LFs by reducing water infiltration, soil moisture availability, aeration, and rooting space, and by increasing the physical resistance for plant root growth which result in increased recruitment difficulty [133–135]. Unlike wide LFs, the surface layer of narrow LFs consists of material excavated from ditches, and a thin layer of construction material aggregates. The poor physical condition of the surface layer and low bearing capacity interfere with narrow LFs' intensity of use [6,127,130]. Hence, it is very likely that the integration of LF clearing width captured underlying differences in hydrological conditions such as water and nutrient availability, driven by compaction and construction substrate type. Additionally, due to uneven vehicular activities, different traffic intensity patterns on wide and narrow LFs likely explain variation in vegetation cover levels between forest road types.

### 6.2.3. Clearing width and Its Relationship to Local Environmental Conditions

The advanced vegetation cover levels on narrow-width forest roads can be attributed to a combination of limited disturbance and favorable growing conditions. Our data support that a range of vegetation covers can be observed, depending on variations in incident sunlight, shade, and wind conditions. Evidence on wind and incident sunlight patterns on LFs come from Stern et al. [22], where LF openings exhibited double incident sunlight intensity and double maximum wind speed compared to the adjacent forests. The abiotic conditions were different between LFs with different clearing widths: wide LFs exhibited increased sunlight penetration that extended into the forest. Centers of wide seismic lines were characterized by >1.5 times higher sunlight intensity than those of narrow seismic lines. These results corroborate the findings in Franklin et al. [21], showing that the microclimatic conditions in the middle of LFs were generally intermediate between the interior forest and anthropogenic infrastructures, such as well pads, with narrow seismic lines more similar to the interior forest and wide seismic lines more similar to well pads. The width and orientation of LFs also influenced growth trends, as shown in Franklin et al. [21], by changing the abiotic environment: regeneration density on seismic lines increased by 5.8 times for each 10-fold increase in sunlight intensity. Our findings showed that wide forest roads experienced lower vegetation cover levels compared to narrow forest roads. Sunlight was a limiting factor and higher wind speed promoted higher levels of vegetation development. These results are not contrary to the findings in Franklin et al. [21], as their sampled wide LFs were older than the narrow LFs and therefore had more time for tree establishment and growth. Moreover, given the ranking of our factors conditioning vegetation cover, it is very likely that the clearing width moderates the changes in abiotic conditions leading to significant variations in vegetation cover levels between

forest road types. Conceptually, clearing width influence various processes: on wide LFs, greater sunlight availability could result in higher temperature and lower moisture levels (warmer and drier conditions near the ground on wide lines) [9,21]. On narrow LFs, however, significant shading from the adjacent canopy provides more favorable conditions for vegetation cover. This supports the assumption that the clearing width is a modulator of online abiotic conditions including sunlight, wind, and moisture [28,136]. Hence, research on the abiotic environment within LFs is needed to provide insight into potential explanations for abiotic–biotic associated patterns. Additionally, the floristic aspect of online communities should be considered for an integrative investigation of vegetation characteristics within LFs [27,137]. Forest roads with low NDVI levels exhibited limited vegetation cover, likely because low NDVI values indicate less or no vegetation. Contrary to van Rensen et al. [28], YPC was among the most influential factors, and it is possible that our continuous factor better accounted for the variation in vegetation cover. Steeply sloped forest roads (i.e., slopes greater than 15%) experienced advanced vegetation cover. A likely explanation for this is that steeper slopes provide favorable subsurface water exchanges and flow, which promote drier terrain conditions. This is supported by the TWI data indicating that increased water accumulation reduces vegetation cover on forest roads.

### 6.3. Characterization of Vegetation Cover Dynamic around Forest Roads

Our model predictions showed that for extended timeframes (>2 decades post-clearing), vegetation cover sustained an overall upward trend; however, slight variations occurred between wide and narrow forest roads, meeting our expectations of a more advanced cover on narrow-width forest roads. Early studies assessing vegetation cover were carried out in Latin America [138,139], South East Asia [51,134] and Central Africa [140]. They provided evidence of the increased disturbance on wide LFs, as well as variations in density, diversity, and vegetation structure across the LF surface and their proximal environments (edge and adjacent forest). These results and findings in Lee and Boutin [27], allowed us to compare our results with respect to the factors associated with vegetation growth and further confirm that disturbance legacies on wide LFs can persist for decades in boreal forests. A characterization of post-clearing vegetation growth patterns within LFs across the range of forest ecosystems is still in development and different definitions of vegetation growth have been proposed in the forest and LF literature (e.g., spectral indices [141], structure: closure through both height (regeneration), and lateral growth [142–144]). These notable limitations in previous studies and data availability over long timeframes constrained our quantitative analysis. Our ability to compare vegetation cover predictions was further constrained by the small number of studies available: many individual studies have not been conducted over the longer timeframes necessary to detect vegetation growth, or growth has not been properly defined to efficiently compare patterns across forest ecosystems, or across different forest regions in Canada [29]. A quantitative study in a Central African forest [145] demonstrated the potential for vegetation growth on abandoned LFs (logging roads) through natural processes: for an average of a 20 m clearing width, twenty-five years following abandonment, canopy closure recovered to 83% (very close to the value in the adjacent forest in their study area). In our study, wide forest roads showed an average vegetation cover of ~3%–53% and ~14%–52% for the mid- ([10–20] YPC) and long-term (>20 YPC) timeframes, respectively. Narrow forest roads showed an average cover of 17%–51% and 40%–82%, in the mid- and long-term, respectively. The differences could be attributed to forest ecosystem specifications (e.g., vegetation and soil conditions), the metric used to quantify vegetation characteristics on the roads, or road construction specifications (e.g., clearing widths). Findings in Lee and Boutin [27], for the boreal forest ecosystem showed low woody vegetation growth increments thirty-five years post-clearing: most LFs in the study (i.e., ~65% of total LFs) remained in a cleared state with a cover of low forbs, and only 8.2% of LFs across all forest types had exhibited more than 50% woody vegetation growth. LF vegetation predictions in Finnegan et al. [125], showed a 1–2 m

height growth increment 10 years post-clearing, with low lateral cover, and it was mostly disturbance-tolerant taxa. Further evidence comes from Revel et al. [146], where the growth increment of saplings was low with most saplings less than 2 m tall 10 years post-clearing. These quantitative measures for LFs highlight the importance of a unified protocol for the study of vegetation growth within LFs, which better standardize the spatiotemporal component to allow for comparisons. This would require the establishment of a coordinated long-term network of monitoring sites within the existing LF network. Moreover, the use of LiDAR data to estimate post-clearing growth patterns would be more straightforward if LFs were stratified by number of years/decades post-clearing. This would help integrate more structure into the sampling scheme and compensate for the large extent of the road network which can make the monitoring task difficult. The examination of growth patterns following fire or harvest in plot-level studies across forest ecosystems showed variable annual increments [29]. The timeframe is five years for cleared areas to attain a benchmark canopy cover of 10% post-fire, compared to 10 years to attain 10% of canopy cover post-harvest. Furthermore, Senf et al. [30], provided a direct quantification of post-clearing vegetation growth increments; the average is 84% of the disturbed areas reaching recovery benchmarks (i.e., a minimum tree cover of 40% and minimum stand height of 5 m), 30 years post-clearing. While comparisons with post-harvest and post-fire growth increments allow us to contextualize and evaluate our findings, some key differences should be noted. For example, linear (e.g., forest roads) and polygonal (e.g., cutblocks) openings differ with respect to spatial footprint, canopy clearing technique, and disturbance legacies.

*6.4. Research Limitations*

The prediction accuracy of the *rf* approach can benefit from the inclusion of additional factors such as transport flux, compaction levels, and specifications on the construction materials. From the comparison results, ensemble approaches such as *rf* and *gbm* showed low error rates. However, additional model calibration and testing are needed to further validate these findings and evaluate the generalization capabilities of these approaches. Additionally, other techniques for factor importance and ML interpretation should also be tested. Similar to the proximity-based analysis in Abib et al., 2019 [1], both cross-validated and independently validated *rf* results satisfied the accuracy and goodness-of-fit criteria. Since repeated measurements provide additional information, it is important that dependencies in the input data are accounted for. For this purpose, stratified random sampling is used when there are strata that need to be considered in the analysis: it reproduces characteristics in the samples that are representative of the strata. Estimates generated within strata are more accurate than those from random sampling because dividing the input data into homogeneous strata often reduces sampling error and increases precision. Nonetheless, we suggest that spatial autocorrelation should be a factor of further analysis in this spatial application. Future studies could further assess model performance in the context of clustered data [147]. In general, the main disadvantages with ML approaches compared to OLS approaches are: (i) simple linear functions are highly approximated; (ii) for certain data sets, it is difficult to constrain the model by selecting the optimum parameters through cross-validation; and (iii) the output can be unstable, for example, small changes in data can produce highly divergent trees for example [119]. In this study, ML approaches, compared to OLS approaches yielded satisfactory accuracy results for the prediction of vegetation dynamics, but there are limitations concerning the generalization of the results of this study. The models were calibrated and tested with samples collected from a range of forest road sizes (i.e., clearing width) and over a bounded years post-clearing interval. Moreover, the samples were taken from three study areas which share common soil and climatic properties. This means that the predictive models could not be generalized for the prediction of the same characteristics in any unsampled location or within-forest roads with different specifications. Because large-area generalization (e.g., regional, national) depends on the variability of the training and test samples, more observations are needed. This would require a greater range of geoclimatic

conditions within forest roads as well as a higher diversity of forest road specifications. Our findings are consistent with recent LiDAR-based studies in the boreal region which have shown that the post-clearing vegetation dynamic is complex and growth increments are low. Our long-term predictions suggest that a timeframe of no less than 20 years must be expected for both wide and narrow LFs to exhibit ~50% and ~80% of vegetation cover, respectively. Future studies could compare growth patterns and evaluate whether the differences between polygonal features (resulting from fire and harvest) and LFs lead towards distinct successional trajectories [133,148]. Another consideration can emerge from this comparison and is related to the linear aspect of anthropogenic infrastructures which makes the application of chrono-sequence approaches difficult [149]. In our analysis, our plots represent points along a spatial continuum; however, the temporal component was constrained to specific data points in time. Therefore, it is important to predict post-clearing growth patterns along a temporal continuum.

## 7. Conclusions

In this study, we characterized within-forest road vegetation cover dynamics for boreal forest ecosystems using LiDAR-based CHM data and predictive modelling. Our predictive accuracy findings demonstrated that the ML approaches performed better than OLS approaches, with the *rf* model providing a better fit over that obtained with other OLS and ML models (RMSE ranging from 18.69% to 20.29% and $R^2$ ranging from 0.69 to 0.62, using stratified cross-validation and independent datasets, respectively). The *rf* model was closely followed by *gbm*, which suggests that tree-based ensemble approaches can improve prediction accuracy. The inability of OLS approaches to handle non-linear relationships between the vegetation cover response and the causal factors is the main limitation for an accurate characterization of forest road vegetation cover dynamics. Clearing width was found to be the most important factor and was followed by years post-clearing, NDVI, shade, and climatic variables in predicting vegetation cover at a fine scale. Vegetation cover varied by forest road type, with narrow-width roads having higher mean vegetation cover predictions (~17%–51% and ~40%–82% across all five buffers extending from the road centerline, for the mid- and long-term timeframes, respectively) compared to wide roads (~3%–53% and ~14%–52% across all five buffers extending from the road centerline, for the mid- and long-term timeframes, respectively). The *rf* prediction capability, though satisfactory, requires further testing for large-area generalization. Additionally, transport flux and volumes, compaction levels, and the construction materials are among the potential factors that could be included to evaluate possible decreases in model error. With the increasing availability of remote sensing datasets, there is potential for broad-scale mapping of vegetation dynamics around forest roads (landscape or regional level). Further investigations are also required to improve the temporal resolution of vegetation measurements with LiDAR.

**Supplementary Materials:** The following are available online at https://www.mdpi.com/article/10.3390/f14030511/s1, Figure S1: Field inventory plot design used to reconstruct the centerlines (Dimensions: 50 m*clearing width). Table S1: Hyperparameters (ranges and types) and their definitions. Figure S2: (A) Summary of vegetation cover predictions (means and means +/- standard deviation error bars) grouped by different forest road categories and timeframes, from cross-validated *rf* model ($R^2$ = 0.69, RMSE = 18.69%) recorded within the multi-buffers around the road centerlines, across forest road types (wide roads and narrow forest roads) for the post-clearing timeframes: >20 YPC (long-term, black boxes), [10–20] YPC (mid-term, dark grey boxes), and [0–10] YPC (short-term, light grey boxes). (B) Vegetation cover mean predictions using independently-validated *rf* model ($R^2$ = 0.62, RMSE = 20.29%) across forest road types and post-clearing timeframes. Figure S3: *rf*-based Partial dependence plots (black curves) showing impacts of single factor on vegetation cover when all remaining factors are constant. Smooth curves are shown in blue.

**Author Contributions:** Conceptualization, N.B. and O.V.; methodology, N.B., O.V. and L.I.; software, N.B. and O.V.; validation, N.B., O.V. and L.I.; formal analysis, N.B. and O.V.; investigation, N.B., O.V. and L.I.; resources, O.V.; data curation, N.B. and O.V.; writing—original draft preparation, N.B. and O.V.; writing—review and editing, N.B., O.V. and L.I.; visualization, N.B., O.V. and L.I.; supervision, O.V. and L.I.; project administration, O.V.; funding acquisition, O.V. All authors have read and agreed to the published version of the manuscript.

**Funding:** This research was funded by Natural Sciences and Engineering Research Council of Canada (NSERC), grant number RDCPJ 543921-19 under the project "Subvention de recherche et développement coopérative—projet (RDCPJ)" "Analyse de la végétalisation des chemins forestiers et de leur utilisation par les prédateurs et compétiteurs du caribou des bois dans le nord du Québec", "avec Eacom Timber Corporation, Matériaux innovants Rayonier, Ministère des Forêts, de la Faune et des Parcs" with the collaboration of Pierre Drapeau as PI for the funding.

**Data Availability Statement:** The data presented in this study are openly available in https://depositum.uqat.ca/ (accessed on 1 January 2023).

**Acknowledgments:** We thank Maxence Martin for his valuable help in the statistical analysis and Thomas Maxime for his help during the field campaign. We thank all the individuals who have been involved in the project for their expertise and assistance in all aspects of our study and for their help in developing and reviewing the manuscript.

**Conflicts of Interest:** The authors declare no conflict of interest.

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
