# Peer review of "Characterization of Vegetation Dynamics on Linear Features Using Airborne Laser Scanning and Ensemble Learning"

_forests, doi:10.3390/f14030511_

Round 1
Reviewer 1 Report
The manuscript by Braham et al. (forests-2176075) identified the vegetation characteristics dynamic on linear features using ordinary least squares (OLS) and machine learning (ML) regression approaches. Then, they investigated the underlying driving mechanism of the vegetation cover dynamic using random forest approach. The methodology is new, and the findings of the paper are interesting. Overall, this manuscript is informative. I provide some comments below, which may be helpful for the authors to further improve the paper.
Specific comment:
L55, 104, 120. LFs mean linear features? Please give an explanation of the abbreviation when first used (LF, CHM, and ML).
L193. In section 3.2, the author uses a very long paragraph to describe the data used. Please use several short paragraphs to describe different types of data, which can make the article more readable
L250. This Figure should be “Figure 2”.
L254.in section 4.1, please use several short paragraphs to describe 6 different approaches.
L256-321. OLS and machine learning approaches were introduced by the authors in detail. However, I would suggest the authors further describe how these approaches are used in this study. For example, what are the prediction and predictors, and how did you derive the relative importance of the predictors? Please specify in the Method section.
L371. Please improve the figure resolution.
L463, 483, 515 Use the template for the numbering of the subsections. Such as 6.2.1., 6.2.2., 6.2.3.
Please check all the figure captions. The figure number is very confused
The quality of the figures should be greatly improved.
Author Response
Response to reviewers
Subject: Submission of the revised version of manuscript entitled “Characterization of vegetation dynamics on linear features using airborne laser scanning and ensemble learning”
Dear Editorial Team,
Please find enclosed our revision of the manuscript entitled “Characterization of vegetation dynamics on linear features using airborne laser scanning and ensemble learning”. We are very grateful for the useful comments and suggestions received from the three reviewers, as well as the editorial team, which have helped to improve the quality of the manuscript considerably. Detailed responses (underlined and indicated using a “**”) to reviewers’ comments are provided below. We are confident that the manuscript will now be acceptable for publication. Please let me know if additional adjustments are required.
Best regards,
Narimene Braham & collaborators
Reviewer 1
The manuscript by Braham et al. (forests-2176075) identified the vegetation characteristics dynamic on linear features using ordinary least squares (OLS) and machine learning (ML) regression approaches. Then, they investigated the underlying driving mechanism of the vegetation cover dynamic using random forest approach. The methodology is new, and the findings of the paper are interesting. Overall, this manuscript is informative. I provide some comments below, which may be helpful for the authors to further improve the paper.
Specific comment:
L55, 104, 120. LFs mean linear features? Please give an explanation of the abbreviation when first used (LF, CHM, and ML).
**The definition had been added the first time these terms had been introduced in the text as recommended.
L193. In section 3.2, the author uses a very long paragraph to describe the data used. Please use several short paragraphs to describe different types of data, which can make the article more readable.
**Section 3.2 had been split as suggested in 2 steps, buffering and extraction (subsections 3.2.1 and 3.2.2), the paragraphs in each step had been separated. I also added a table for introducing the data used in the analysis. (The only reason the table had not been added from the beginning, is because we think that a repetition of the text to describe the data).
L250. This Figure should be “Figure 2”.
**Figure annotations had been edited.
L254.in section 4.1, please use several short paragraphs to describe 6 different approaches.
**The text was separated into separate paragraphs as requested.
L256-321. OLS and machine learning approaches were introduced by the authors in detail. However, I would suggest the authors further describe how these approaches are used in this study. For example, what are the prediction and predictors, and how did you derive the relative importance of the predictors? Please specify in the Method section.
**Since we didn’t have high-dimensional data input to train our models, all the factors were used for the ML models. For the non-ML models, I mention that we screen for non-informative factors using Mumin (Line 362 to 366).
“The parsimony of mlr and gam approaches were assessed with the Akaike information criterion (AIC) [95]. All possible combinations of factors and interaction effects were analyzed with the MuMIn library in R [96]. This step was essential because the inclusion of uninformative factors in parametric and semi-parametric models (i.e., mlr and gam) can reduce their overall predictive performance.”
Should the information about the factors be included in the table that I added?
L371. Please improve the figure resolution.
**Figures had been tweaked to enhance the quality of the text, and to homogenize with the template’s fonts.
L463, 483, 515 Use the template for the numbering of the subsections. Such as 6.2.1., 6.2.2., 6.2.3.
**Renamed as indicated in the template.
Please check all the figure captions. The figure number is very confused
**Figures annotations checked.
The quality of the figures should be greatly improved.
**Figures checked and improved as requested.

Reviewer 2 Report
The article deals with the very topical topic of forest restoration in the vicinity of forest roads used for access to the forest and for the movement of logging machines.
In the Introduction chapter, the existing knowledge is sufficiently described and relevant literature is cited. Unfortunately, there is no clear statement of the objectives of the thesis, which must be completed.
Although the methodology is described quite well from my point of view, it is not clearly explained at first glance that the plots are located directly on forest roads, it would be useful to mention a reference to Figure 1 in Chapter 3.1. In addition, Figure.1 is mentioned 2 times, so I ask to renumber the figures.
In chapter 3.2 the use of LiDAR data is described, but the parameters of Airborne Laser Scanning are not given, although the density of points is mentioned, but I wonder if the data was collected during the growing season or outside. The imaging period can have a significant effect on the resulting canopy surface model. Line 217 does not explain the creation of the CHM. I don´t feel qualified to judge English language but more suitable than Northerness is Aspect.
The accuracy of climate characteristics prediction in the WorldClim model depends on the number and distribution of climate stations in the area, so the data may not always be completely accurate, are there really no better data for the area? Moreover, given the resolution of the data, very little variation can be expected within individual study areas.
I was not able to find exactly how the incident solar radiation was calculated in the WorldClim model, but in the case of LiDAR the data would be much more accurate for individual plots calculated in a GIS environment.
In chapter 4. Methods are only mentioned abbreviations for different ML regression and are explained only below Figure 3., please describe in the text.
In the evaluation of the results, the Norterness factor came out as highly insignificant, why was it not removed from the analysis?
Also, I'm not sure if it can be argued from the results that incident solar radiation and TWI has an effect on vegetation growth. While it certainly does, it is not entirely obvious from the results of the analysis.
Although most of the comments are really only marginal, I am returning to the major revision primarily for clarity of objectives.
Author Response
Response to reviewers
Subject: Submission of the revised version of manuscript entitled “Characterization of vegetation dynamics on linear features using airborne laser scanning and ensemble learning”
Dear Editorial Team,
Please find enclosed our revision of the manuscript entitled “Characterization of vegetation dynamics on linear features using airborne laser scanning and ensemble learning”. We are very grateful for the useful comments and suggestions received from the three reviewers, as well as the editorial team, which have helped to improve the quality of the manuscript considerably. Detailed responses (underlined and indicated using a “**”) to reviewers’ comments are provided below. We are confident that the manuscript will now be acceptable for publication. Please let me know if additional adjustments are required.
Best regards,
Narimene Braham & collaborators
Reviewer 2
The article deals with the very topical topic of forest restoration in the vicinity of forest roads used for access to the forest and for the movement of logging machines.
In the Introduction chapter, the existing knowledge is sufficiently described and relevant literature is cited. Unfortunately, there is no clear statement of the objectives of the thesis, which must be completed.
**The objectives and hypotheses paragraph is highlighted in the 1.Introduction (Line 146 to 150)
** we thought the specific objectives were clearly stated in the text, however, here is a variation of how it can be stated (Let us know if it is necessary to re-formulate),
“-The main aim of the study is to develop a modelling framework for a fine-scale characterization of vegetation conditions within forest roads, using 1 m spatial resolution LiDAR data, along with additional geospatial information.
-Specific research objectives of the study are:
- The evaluation of the relevance of different modelling approaches, to investigate their performance.
- The assessment of the relative importance of the different factors, driving the prediction of vegetation cover.”
Although the methodology is described quite well from my point of view, it is not clearly explained at first glance that the plots are located directly on forest roads, it would be useful to mention a reference to Figure 1 in Chapter 3.1. In addition, Figure.1 is mentioned 2 times, so I ask to renumber the figures.
**Reference to the plots in Chapter 3.1. added (Line 210 to 213) .
**Figure annotation had been edited.
In chapter 3.2 the use of LiDAR data is described, but the parameters of Airborne Laser Scanning are not given, although the density of points is mentioned, but I wonder if the data was collected during the growing season or outside. The imaging period can have a significant effect on the resulting canopy surface model. Line 217 does not explain the creation of the CHM. I don´t feel qualified to judge English language but more suitable than Northerness is Aspect.
**Growing season information was added (leaf-on conditions). We add the available information about the flights specifications and refer to the Quebec government web site for additional info (Line 242)
**A description of LiDAR products and extraction of CHM added (222 to 235).
** Aspect is more general, that is why Northerness was used.
The accuracy of climate characteristics prediction in the WorldClim model depends on the number and distribution of climate stations in the area, so the data may not always be completely accurate, are there really no better data for the area? Moreover, given the resolution of the data, very little variation can be expected within individual study areas.
I was not able to find exactly how the incident solar radiation was calculated in the WorldClim model, but in the case of LiDAR the data would be much more accurate for individual plots calculated in a GIS environment.
** This makes a great point. At some point I tried to extract the topographic incident solar radiation from LiDAR, using different GIS software, without success.
In chapter 4. Methods are only mentioned abbreviations for different ML regression and are explained only below Figure 3., please describe in the text.
**I think abbreviations were first introduced in the Introduction (starting Line 121).
In the evaluation of the results, the Norterness factor came out as highly insignificant, why was it not removed from the analysis?
**We run the ML models using the same set of predictors, all at once. For all ML methods, all predictors were included. The predictors were reduced for the MLR and gam methods.
Also, I'm not sure if it can be argued from the results that incident solar radiation and TWI has an effect on vegetation growth. While it certainly does, it is not entirely obvious from the results of the analysis. Although most of the comments are really only marginal, I am returning to the major revision primarily for clarity of objectives.
-It could be suggested that:
1.different statistics should be tested (min, max, mean, or sd) for the extraction of the variables to assess their effect (for the Northerness variable)
- Expand the range of values for the variables that ranked last.
It is possible to explore these two options, however this does not present a particular incidence on our results.

Reviewer 3 Report
Manuscript:
Characterization of vegetation dynamics on linear features using airborne laser scanning and ensemble learning
submitted by:
Narimene Braham, Osvaldo Valeria and Louis Imbeau
Linear features network developed throughout much of the commercial boreal forest. They also have environmental implications which involve both the active and non-active portions of the network. Management of the existing linear features network across the boreal forest would lead to the optimization of maintenance and construction costs as well as the minimization of the cumulative environmental effects of the anthropogenic linear footprint. The study has improved Authors' understanding of fine-scale vegetation dynamic within forest roads and demonstrates that spatially-explicit models using LiDAR data are reliable tools for assessing vegetation dynamics within forest roads. In Authors' opinion, improved knowledge of vegetation dynamic patterns on linear features can help support sustainable forest management.
The structure of the manuscript is considered and clear. In the introduction, the background and comprehensive review of the problem's literature were presented. The Authors present statistical approaches, models parameter tuning, performance, comparison, and diagnostics using cross-validation and independent dataset. Results of the research have been presented in tabular and graphic form. Discussion is very deep and conclusions, on the basis of the research, are comprehensive and clear.
Following suggestions should be taken into consideration:
Line 62: [37]–, is unclear, please check
Table 1: coordinates should be positive; -77.23°W should be replaced by 77.23°W in whole the manuscript
Line 233: units in following form (kJ m-2 day-1)are unclear, I suggest: (kJ×m-2×day-1)
Line 321: I have not found Supplementary material neither in the manuscript nor separate file
Line 358: as line 233
Table 2 can be adjusted to left margin, then first column is wider
Line 412: left bracket is probably wrong ]10–20], please check
Line 425: as above
Line 658: probably R2 should be raplaced by R2 - please check
Table footers should be below the table, not its part
References should be adjusted to instruction fo Authors' requirements
Author Response
Response to reviewers
Subject: Submission of the revised version of manuscript entitled “Characterization of vegetation dynamics on linear features using airborne laser scanning and ensemble learning”
Dear Editorial Team,
Please find enclosed our revision of the manuscript entitled “Characterization of vegetation dynamics on linear features using airborne laser scanning and ensemble learning”. We are very grateful for the useful comments and suggestions received from the three reviewers, as well as the editorial team, which have helped to improve the quality of the manuscript considerably. Detailed responses (underlined and indicated using a “**”) to reviewers’ comments are provided below. We are confident that the manuscript will now be acceptable for publication. Please let me know if additional adjustments are required.
Best regards,
Narimene Braham & collaborators
Review report 3
Manuscript: Characterization of vegetation dynamics on linear features using airborne laser scanning and ensemble learning
submitted by: Narimene Braham, Osvaldo Valeria and Louis Imbeau
Linear features network developed throughout much of the commercial boreal forest. They also have environmental implications which involve both the active and non-active portions of the network. Management of the existing linear features network across the boreal forest would lead to the optimization of maintenance and construction costs as well as the minimization of the cumulative environmental effects of the anthropogenic linear footprint. The study has improved Authors' understanding of fine-scale vegetation dynamic within forest roads and demonstrates that spatially-explicit models using LiDAR data are reliable tools for assessing vegetation dynamics within forest roads. In Authors' opinion, improved knowledge of vegetation dynamic patterns on linear features can help support sustainable forest management.
The structure of the manuscript is considered and clear. In the introduction, the background and comprehensive review of the problem's literature were presented. The Authors present statistical approaches, models parameter tuning, performance, comparison, and diagnostics using cross-validation and independent dataset. Results of the research have been presented in tabular and graphic form. Discussion is very deep and conclusions, on the basis of the research, are comprehensive and clear.
**We thank the reviewer for this description about the contribution of our work.
Following suggestions should be taken into consideration:
Line 62: [37]–, is unclear, please check
**Edited (Line 164-165)
Table 1: coordinates should be positive; -77.23°W should be replaced by 77.23°W in whole the manuscript
**Coordinates Edited, thank you for this edit.
Line 233: units in following form (kJ m-2 day-1) are unclear, I suggest: (kJ×m-2×day-1)
**Edited (I added points () instead of the crosses). I also edited (), as suggested.
Line 321: I have not found Supplementary material neither in the manuscript nor separate file
**Supplementary Materials was the last section of the manuscript when it was submitted (Figure S1 to S4) (?).
Line 358: as line 233
*Checked factors names throughout the text.
Table 2 can be adjusted to left margin, then first column is wider
**Edited
Line 412: left bracket is probably wrong ]10–20], please check
*It is just to say that the 10-year value is not included in the interval.
Line 425: as above
Line 658: probably R2 should be raplaced by R2 - please check
**Edited.
Table footers should be below the table, not its part
** I fixed the footer.
References should be adjusted to instruction for Authors' requirements.

Round 2
Reviewer 2 Report
Dear authors,
thank you for the addition and correction. I am now convinced that the article is suitable for publication in Forests.